# Standardized Incidence Ratio and Standardized Mortality Ratio of Malignant Mesothelioma in a Worker Cohort Using Employment Insurance Database in Korea

**DOI:** 10.3390/ijerph182010682

**Published:** 2021-10-12

**Authors:** Eun-A Kim

**Affiliations:** Occupational Safety and Health Research Institute, Korea Occupational Safety and Health Agency, Ulsan 44429, Korea; toxneuro@kosha.or.kr; Tel.: +82-52-7030-0810

**Keywords:** mesothelioma, cohort, asbestos, Korea, incidence ration, mortality ratio

## Abstract

Malignant mesothelioma is one of the appropriate indicators for assessing the carcinogenic effects of asbestos. This study compared the risk ratio of mesothelioma according to the industry in the worker cohort. A cohort was constructed using the Korean employment insurance system during 1995–2017, enrolling 13,285,895 men and 10,452,705 women. The standardized mortality ratio (SMR) and standardized incidence ratio (SIR) were calculated using the indirect standardization method. There were 641 malignant mesotheliomas that occurred; the SIR was significantly higher than the general population (men 1.36, 95% confidence interval (CI) 1.24–1.48, women 1.44, 95% CI: 1.23–1.7). More than half (52.8%) of malignant mesothelioma cases occurred in the manufacturing (*n* = 240, 38.6%, SIR: men, 1.72, 95% CI: 1.37–2.15, women, 3.31, 95% CI: 1.71–5.79) and construction industries (*n* = 88, 14.2%, SIR: men, 1.54 95% CI: 1.33–1.78, women, 1.62 95% CI: 1.25–2.11). The accommodation and food service (men, 2.56 95% CI: 1.28–4.58, women 1.35, 95% CI: 0.65–2.48) and real estate (men 1.34, 95% CI: 0.98–1.83, women 1.95, 95% CI: 0.78–4.02) also showed a high SIR, indicating the risk of asbestos-containing materials in old buildings. The incidence of malignant mesothelioma is likely to increase in the future, given the long latency of this disease. Moreover, long-term follow-up studies will be needed.

## 1. Introduction

Asbestos-related diseases have a long latency period, limiting accurate identification of past exposure. Like most occupational cancers [1], malignant mesothelioma has a long latency period, 40 years or more. The longer the latent period, the more difficult it is to find information on carcinogen exposure, making it difficult to determine its occupational relevance. Fortunately, malignant mesothelioma is called a signal tumor [2], and it is relatively easy to determine the occupational causality because it has a high occupational attributable fraction. Unlike other asbestos-related diseases, such as lung cancer and laryngeal cancer, malignant mesothelioma is almost unaffected by smoking [3], so it can be used as an appropriate indicator to examine the health effects of asbestos.

The use of asbestos in Korea began with asbestos mining in the 1930s until World War II. From the 1960s, the asbestos mining industry was restarted by the asbestos cement industry. Most asbestos production was chrysotile with small amounts of tremolite and sepiolite [4]. The major use of asbestos in Korea was in construction materials, such as slate roof tiles. In 1997, crocidolite and amosite were banned, before all forms of asbestos were prohibited in Korea in 2009 [5].

The incidence of malignant mesothelioma in Korea was 0.3 per 100,000 as of 2018 [6], making it a rare disease. Calculations of the standardized mortality ratios (SMR) or standardized incidence ratios (SIR) for low-incidence cancers require the establishment of large cohorts of workers. Thus far, studies on malignant mesothelioma in Korea have mainly been conducted as studies related to environmental exposure [7,8,9,10]. Many studies of workers’ malignant mesothelioma analyzed workers’ compensation cases [11].

Various industries have a high risk of asbestos exposure: petroleum refining [12]; pharmaceuticals [13]; chemicals [14]; rubber and plastics [15]; non-metallic products, basic metal manufacturing, metal processing product manufacturing, machinery, and equipment manufacturing [14,16]; electronic parts and electrical equipment manufacturing [17]; shipbuilding [18,19]. In Korea, malignant mesothelioma was reported from the construction [20], shipbuilding, and asbestos textile production industries [21]. On the other hand, these studies were conducted as case-control studies [22] or analyzed as case series reported from pathology laboratories [20]. Kang et al. [23] reported that various industries, including mining, textile industry, iron and steel industry, automobile repair services, paper box, or container manufacturing industry, can be classified as asbestos-related industries in Korea. Therefore, this study compared the risk ratios of malignant mesothelioma according to the industries in the worker cohort compared to the general population.

## 2. Materials and Methods

A cohort was constructed using the Korean employment insurance system. All workers who signed up for employment insurance from 1 July 1995 to 31 December 2017 were enrolled. The exclusion criteria were cases who developed malignant mesothelioma or died before the initial enrollment in employment insurance. The employment insurance system database contains information on the enrollment and withdrawal dates, industries, and occupations. The system covers 48.5% of all workers because the employment insurance system excludes public officials, military personnel, and those eligible for private education pensions and participation in household employment activities or small construction businesses below a certain standard.

The final cohort enrolled 13,285,895 men and 10,452,705 women. The occurrence and death from malignant mesothelioma were confirmed by linking the database of the cancer registration center and the death registration database of the national statistical office. The databases were linked by the personal resident registration number.

Malignant mesothelioma was limited to cases diagnosed with the international standard disease classification code C45. The histological diagnosis rate of the cancer registration center is managed at a level of 93%.

The industry classification of the workers was based on the longest workplaces where each worker had worked. Because many workers worked in more than one industry during the follow-up period, there might be some misclassification of the workers’ asbestos exposure risk. On the other hand, to evaluate each specific industry, the workers need to be classified according to their major working industry.

The industry classification was based on the 10th Korea Standard Industry Code (KSIC) [6]. The KSIC was based on the International Standard Industrial Classification (ISIC) adopted by the UN, divided into 21 major classifications. The major classifications can be defined by the subsequent breakdowns to the five digits.

The standardized mortality ratio and standardized incidence ratio of the cohort were calculated using the indirect standardization method. The mortality rate of the general population was calculated using the number of deaths in the year obtained through the microdata system [24] of the national statistical office and the national resident registration statistics of the national statistical office [6]. The cancer incidence rate of the general population was calculated using the number of cancer incidences in the year that was collected by the cancer registration center and published by the national statistical office [6]. The SIR and SMR were calculated using the KSICs subdivisions.

## 3. Results

### 3.1. General Characteristics of the Subjects

There were more men in the cohort (56% men and 44% women). Owing to the nature of employment insurance, 75% of the survey subjects were in their 20s and 50s at the cohort enrollment. Those who registered under the age of 20 were men (3.5%) and women (4.8%), and those enrolled over 50 years old were 21.2% men and 20% women. Approximately 54.5% of men and 61.3% of women signed up for employment insurance after 2011, and between 2001 and 2010, 29.3% of men and 28.6% of women were insured. Approximately 16.1% of men and 20.0% of women subscribed before 2000.

The average follow-up period of the cohort was eight–nine years, and the total person-years were 179,932,948. The total deaths from all causes were 534,829, and the SMR was 1.28 (1.28–1.29) for men and 0.95 (0.94–0.95) for women. The incidence and death from malignant mesothelioma were 480 men and 141 women, and 216 men and 54 women, respectively.

Manufacturing was the industry where the workers had worked the longest: 46.7% and 33.3% of men and women of the cohort subjects, respectively. For men, the wholesale and retail industry (10.4%), construction industry (9.3%), transportation and warehousing industry (6.1%), business facility management and business support, and rental service industry (4.0%) were the major industries. For women, the major industries were wholesale and retail business (14.5%), health and welfare business (9.9%), food and lodging business (7.8%), business facility management and business support, and rental service business (6.3%).

An attempt was made to apply the Korean asbestos job-exposure matrix (JEM) [23], having a job and industry dimensions, to this cohort. Only 8.3% of men and 12.5% of women were classified in the asbestos JEM because of the lack of job information. In this study, the longest working industry of cohort members were classified only by the industry corresponding to the asbestos JEM. Approximately 39.6% of men and women were classified as working in the industry specified in the asbestos JEM (Table 1).

### 3.2. SMR, SIR by Major Division of the KSIC

The SIR of malignant mesothelioma in the entire cohort was 1.36 (1.24–1.48) for men and 1.44 (1.23–1.7) for women, which was significantly higher than that of the general population. The SMR was 0.88 (0.77–1.01) for men and 0.95 (0.73–1.24) for women, which was lower than that of the general population, but the difference was not statistically significant.

The industry with the highest incidence of malignant mesothelioma in men was the manufacturing industry (*n* = 183, 1.54 95% CI: 1.33–1.78). This was followed in order by the construction industry (*n* = 76, 1.72, 95% CI: 1.37–2.15); transportation and warehousing industry (*n* = 48, 1.26, 95% CI: 0.95–1.67); real estate industry (*n* = 39, 1.34, 95% CI: 0.98–1.83), business facilities management, business support, and rental service industry (*n* = 22, 0.82, 95% CI: 0.54–1.25); information and communication industry (*n* = 14, 1.7 95% CI: 0.93–2.85); and accommodation and foodservice industry (*n* = 11, 2.56, 95% CI: 1.28–4.58).

The SIR of manufacturing, construction, and lodging, and catering industries were statistically significantly higher than the general population.

Among women, the manufacturing industry had the highest incidence (*n* = 52, 1.62, 95% CI: 1.25–2.11). This was followed in order by the construction industry (*n* = 12, 3.31, 95% CI: 1.71–5.79); wholesale and retail industry (*n* = 12, 1.31, 95% CI: 0.69–2.35); business facility management, business support, and rental service industry (*n* = 11, 1.02, 95% CI: 0.51–1.82); and accommodation and catering industry (*n* = 10, 1.35, 95% CI: 0.65–2.48). Among them, the manufacturing and construction industries had a significantly higher SIR than the general population.

The SMR of men in the construction industry was 1.4 (1.04–1.89), which was significantly higher than that of the general population. The men’s SMR that was higher than that of the general population as follows: agriculture, forestry, and fisheries (1.63, 95% CI: 0.66–3.37); manufacturing (1.02, 95% CI: 0.82–1.27); lodging and food industries (2.07, 95% CI: 0.76–4.51); and information and communication industries (1.29, 95% CI: 0.52–2.65). Nevertheless, the difference was not significant. The women’s SMR that was higher than that of the general population as follows: agriculture, forestry, and fisheries (1.56, 95% CI: 0.04–8.69); manufacturing (1.30, 95% CI: 0.89–1.91); construction (2.36, 95% CI: 0.77–5.51); information and communication industries (1.86, 95% CI: 0.23–6.71); financial insurance industry (1.5, 95% CI: 0.18–5.4); and real estate industry (1.26, 95% CI: 0.26–3.68) (Table 2).

### 3.3. SIR, SMR from Subdivisions of the Manufacturing Industry

Among the manufacturing industries, 14 subdivisions with significantly high SIR and SMR of malignant mesothelioma were as follows: the manufacture of food products (KSIC 10); manufacture of textiles, except for apparel (KSIC 13); manufacture of wood and products of wood and cork, except furniture (KSIC 16); manufacture of pulp, paper, and paper products (KSIC 17); coke, briquettes, and petroleum products manufacturing (KSIC 19); chemicals and chemical manufacturing (KSIC 20); pharmaceuticals manufacturing (KSIC 21); rubber and plastics manufacturing (KSIC 22); non-metallic products manufacturing (KSIC 23); manufacture of basic metals (KSIC 24); manufacture of fabricated metal products, except machinery and furniture (KSIC 25); manufacture of electronic components, computers, visual, sounding, and communication equipment (KSIC 26); electrical equipment manufacturing (KSIC 28); and other machinery and equipment manufacturing (KSIC 29).

In the food manufacturing industry, men worked in the processing and preserving of fish, crustaceans, mollusks, and seaweeds (KSCI 102) (SIR 4.13, 95% CI: 1.12–10.56). Women worked in the processing and preserving of fruit and vegetables (KSCI 103) (SMR 6.31, 95% CI: 1.3–18.45), and in the manufacture of grain mill, starches, and starch products (KSCI 106) (SIR 4.13, 95% CI: 1.12–10.56). SIR 13.27, 95% CI: 1.61–47.94). These areas revealed high risk ratios.

Among the risk ratios for the manufacture of textiles, except apparel (KSIC 13). Men worked in the weaving of cotton fabrics (KSIC 13211) (SIR 8.66, 95% CI: 1.79–25.4, SMR 8.34, 95% CI: 1.01–30.12) and the manufacture of other textiles (KSIC 13999) (men 4.49, 95% CI: 0.93–13.23, women 7.27, 95% CI: 1.5–21.25) showed high ratios.

The manufacture of wood and products of wood (KSIC 16) (SIR 7.31, 95% CI: 1.51–21.36) and manufacture of pulp, paper, and paper products (KSIC 17) (SIR 5.89, 95% CI: 1.6–15.08) were statistically significant and higher in women than men.

Petroleum refining (KSIC 19210) (SIR 6.37, 95% CI: 1.31–18.36), manufacture of other chemical products (KSIC 20499) (SIR 4.28, 95% CI: 1.17–10.95), pharmaceutical manufacturing (KSIC 21) (SIR 3.27, 95% CI: 1.06–7.62), manufacture of plastic cases, boxes and containers (KSIC 22232) (SIR 10.17, 95% CI: 2.1–29.73, SMR 9.96, 95% CI: 1.21–35.99), manufacturing of other structural metal products (KSIC 25119) (SIR 3.78, 95% CI: 1.03–9.67), manufacturing of components (KSIC 26) (SIR 2.12, 95% CI: 1.23–3.39), and electric motors and generators (KSIC 28111) (SMR 8.18, 95% CI: 0.99–25.93) were higher in men than women.

In the manufacture of basic metals (KSIC 24), both men (1.74, 95% CI: 0.93–2.97) and women (3.94, 95% CI: 0.48–14.22) showed high SIRs without statistical significance. The manufacture of other machinery and equipment (KSIC 29) also had a significantly high SIRs in both men (SIR 1.89, 95% CI: 1.1–2.96) and women (SIR 8.61, 95% CI: 1.04–31.12) (Table 3).

### 3.4. SIR and SMR of Non-Manufacturing Industry

Among the general construction industries, the building construction (KSIC 411) in men (SIR 1.63, 95% CI: 1.1–2.43, SMR 1.97 95% CI: 1.28–3.02) and women (SIR 5.86, 95% CI: 2.36–12.08, SMR 5.34, 95% CI: 1.45–13.67) were all significantly higher than the general population. The SIRs of heavy and civil engineering construction (KSIC 412) (2.17, 95% CI: 1.04–3.99), unclassified construction (KSIC 419) (1.69, 95% CI: 1.23–2.33), and construction of installing building equipment (KSIC 422) (6.61, 95% CI: 1.36–19.33) were high in men.

In the wholesale and retail trade sector, the SIR of women in the retail sale of foods, beverages, and tobacco in specialized stores (KSIC 472) (8.67, 95% CI: 1.05–31.31) was high.

The SIR of men in warehousing and support activities for transportation (KSIC 52) (2.05, 95% CI: 1.06–3.57) was also high. The SIR of women was high in the other information service activities (KSIC 63999) (9.92, 95% CI: 1.2–35.83).

The SIR of men in the accommodation industry (KSIC 55) was high (2.56, 95% CI: 1.28–4.58). In particular, the general accommodation and accommodation with cooking facilities (KSIC 551) showed high SMR (4.22, 95% CI: 1.37–9.85).

In residential real estate management (KSIC 68211), the SIR of men was high (1.40, 95% CI: 0.99–2). The administration of industrial and social policy of community (KSIC 842) showed a high SIR in women (6.31, 95% CI: 1.3–18.44) (Table 4).

## 4. Discussion

There were 641 cases of malignant mesothelioma in this cohort. Compared to the general population, the SIR was significantly higher than the general population (men 1.36, 95% CI: 1.24–1.48, women 1.44, 95% CI: 1.23–1.7) (Table 2). More than half (52.8%) of malignant mesothelioma occurred in the manufacturing (*n* = 240, 38.6%) and construction industries (*n* = 88, 14.2%). Many studies reported high risk ratios of malignant mesothelioma from construction and shipbuilding.

According to the results of a nationwide survey of malignant mesothelioma cases in Japan from 2003 to 2008 [25], the construction (131 of 607, 22.6%) and shipbuilding (15.0%) industries had the most mesothelioma cases. In a study from Massachusetts, USA, during 1998–2003 [26], the mesothelioma cases in the construction industry were 22.6% (123 of 543), with a standardized morbidity odds ratio (SMOR) of 3.2 (2.6–3.9). Among the manufacturing industries, the SMORs were highest in ship and boat building and repairing (38.2, 95% CI: 29.4–49.7), and in the pulp, paper, and paperboard mills (2.9, 95% CI: 1.2–7.1). A national survey in the US during 1999–2015 reported high proportionate mortality ratios (PMR) of malignant mesothelioma industries in boat building and repairing (6.7, 95% CI: 4.3–9.9), petroleum refining (4.1, 95% CI: 2.6–6.0), and industrial and miscellaneous chemicals (3.8, 95% CI: 2.9–5.0) [27]. A case-control study of French national surveillance of pleural mesothelioma from 1998 to 2003 reported high odds ratios in the construction industry, ship repair industry, asbestos manufacturing industry, and metal construction material manufacturing industry [28]. In the asbestos JEM of Australia, workers in the asbestos manufacturing, shipyard, and insulation industries were estimated to have had the highest exposure [29]. A Spanish case-control study of pleural mesothelioma, conducted between 1993 and 1996, reported high odds ratios in the manufacture of rubber and plastic products (2.66, 95% CI: 1.11–6.39), manufacture of other non-metallic mineral products (2.23, 95% CI: 1.22–4.09), and manufacture of transport equipment (2.08, 95% CI: 1.08–4.00) [15]. In a case-control study in Italy using the mortality database of Istituto Superiore di Sanità, mesothelioma clusters were found at the near vicinity of the major asbestos-cement plants, naval shipyards, petrochemical plants, refineries, chemical, and textile industries [30].

The textile manufacturing industry is known for its asbestos exposure risk. The present results showed that men in cotton weaving (KSIC 13211) (SIR 8.66, 95% CI: 1.79–25.4, SMR 8.34, 95% CI: 1.01–30.12) and manufacture of other textiles (KSIC 13999) (men 4.49, 95% CI: 0.93–13.23, women 7.27, 95% CI: 1.5–21.25), which is not the asbestos textile industry, showed high ratios. An increased risk of malignant mesothelioma was also reported in the non-asbestos textile manufacturing industry [31]. This can be explained by the use of asbestos for ceilings, pipes, and complementary materials of the machine itself [32].

In this study, women’s SIR was higher than men’s in some industries. In the construction industry, both men and women had a high SIR of malignant mesothelioma. In particular, in the residential building construction industry, the SIR of women (5.86, 95% CI: 2.36–12.08) was higher than men (1.63, 95% CI: 1.1–2.41). Wood products manufacturing, pulp, and paper manufacturing were the main industries using asbestos until the 1970s [33]. In this study, the SIR of women from the manufacture of wood and products of wood (KSIC 16) (SIR 7.31, 95% CI: 1.51–21.36) and the manufacture of pulp, paper, and paper products (KSIC 17) (SIR 5.89, 95% CI: 1.6–15.08) were higher than men (Table 4). A study on the surveillance system for malignant mesothelioma in Italy reported that more women were affected than men in the chemical, plastics manufacturing, and textile manufacturing industries [34]. Additional research will be needed to analyze the exposure level in industries where women showed a higher SIR than men.

Analyses of workers’ mesothelioma cases on a national scale in Korea are rare. In 2012, Jung et al. constructed a monitoring system linked to pathology laboratories across the country. They estimated that 56 cases (36.8%) out of 152 cases were occupational-related cases, and 19.7% of them had a working history in the construction industry [20]. Kang et al. reconstructed the Korean asbestos JEM using previous Korean asbestos JEM [35] and the exposure database in Korea [23]. The industries with high asbestos exposure reported varied according to the era. The authors focused on asbestos mining, asbestos textile industry, asbestos cement industry, iron, and steel industry, ship manufacturing, and sale of motor vehicles and parts in the 1980s; textile and textile manufacturing, wood industry, manufacture of plastics products, asbestos fiber manufacturing, and automobile repair services in the 1990s. In the 2000s, the manufacture of paper, manufacture of paper boxed or containers, synthetic resin or plastic manufacturing industry, automobile paint, and construction stone industry were classified as risky industries. Since the enrollment of the cohort in this study started from 1995, it is expected that the risk of malignant mesothelioma will be high in high risk industries of the 2000s in Kang et al.’s asbestos JEM. On the other hand, these results showed that industries with a high SIR and SMR were distributed evenly among the high exposure industries of 1980s, 1990s, and 2000s in Kang et al.’s asbestos JEM, such as the steel industry, ship manufacturing, textile and textile manufacturing, wood, plastic product manufacturing, paper product manufacturing, and construction (Table 3). The cohort of this study was formed in 1995, and 21.2% of the participants were over 50 years of age at their time of enrollment. Therefore, it is very likely that they had work experience in this workplace prior to 1995 (Table 1).

Retail sale of foods, beverages, and tobacco in specialized stores (KSIC 472), warehousing and support activities for transportation (KSIC 52), information service activities (KSIC 63999), accommodation industry (KSIC 55), management of residential real estate (KSIC 6821), and administration of industrial and social policy of community (KSIC 842) showed significantly high SIRs in this study. These were not emphasized in the asbestos JEM by Kang et al. In this study, a high number of malignant mesothelioma cases were found in the transport and storage (7.9%), real estate activities (7.4%), and accommodation and food service activities (3.4%). Surprisingly, among the major branches of the economy, accommodation and food services were the branches with the highest significant SIR for men (2.56, 95% CI: 1.28–4.58) (Table 2).

Asbestos in Korea produced or used since the 1930s peaked in the mid-1990s and was officially banned in 2009 [7]. Before asbestos was banned, all roofing slate materials were manufactured with a mixture of approximately 90% cement and 10% chrysotile [36]. Therefore, the high SIR of the accommodation, transport and storage, and real estate industry might be related to the exposure to parts of old building materials. The number of malignant mesothelioma cases by year has been increasing in industries, including real estate and accommodation, rather than manufacturing or construction since 2009, when asbestos was banned (Figure 1). In other words, malignant mesothelioma may increase further in the future due to exposure to asbestos-containing substances contained in various materials even now, 10 years after asbestos was banned.

Another possibility is that the malignant mesothelioma patient’s past work experience in asbestos-related occupations may have been overlooked due to the lack of occupational information in this cohort. This possibility requires further review and analysis.

In the cohort constructed in this study, 607 cases of malignant mesothelioma occurred after 1999. This corresponds to 32.9% of the 1846 cases of malignant mesothelioma reported by the cancer registration center in Korea during the same period [6]. Kang et al. (2018) estimated the population attributed fraction (PAF) of asbestos and suggested that 808 malignant mesotheliomas between 1998 and 2013 may have been caused by occupational exposure [37]. The malignant mesothelioma cases of this cohort during the same period were 361 (Figure 1), which is lower than those predicted by Kang et al. This is because the cohort constructed in this study used the employment insurance database. Korea’s employment insurance excluded public officials, military personnel, and those eligible for private education pensions. Hence, participation in household employment activities or small construction businesses below a certain standard is restricted. This database cannot represent the entire workforce of Korea because business owners and self-employed people are stipulated to join voluntarily [38].

Although the latent period of malignant mesothelioma has been reported to be more than 40 years, the average follow-up period of this cohort was eight–nine years, which is not sufficient to determine the tendency of malignant mesothelioma in this cohort. Therefore, additional analysis will be needed through long-term follow-ups on the risk of occurrence and death according to the industry with the longest working period currently analyzed.

Another possibility of bias in this study was the use of the longest working history. The exposure risk of workers who worked in more than one industry might be misclassified. The results of this study might have a systemic bias because the longest working industry is not necessarily the highest exposure industry. Nevertheless, the longest industry was selected for analysis due to the lack of a complete working history of this cohort.

Despite these limitations, this study conducted the first retrospective cohort study with limited data from a Korean industrial workers’ insurance fund (1995–2017). As a result, the risk of malignant mesothelioma was high in previously well-known asbestos-risk industries. In addition, the risk of malignant mesothelioma is high in the lodging and real estate management industries. Therefore, it will be necessary to analyze the entire cohort by occupation to derive more accurate information. This type of study will require cooperation between government units that collect social databases.

## 5. Conclusions

From the results of this study, the SIR of malignant mesothelioma in Korean workers was 1.36 (1.24–1.48) in men and 1.44 (1.23–1.7) in women, which was significantly higher than the general population. The construction industry (men 1.72, 95% CI: 1.37–2.15, women 3.31, 95% CI: 1.71–5.79) or manufacturing sector (men 1.54, 95% CI: 1.33–1.78, women 1.62, 95% CI: 1.25–2.11), which are traditionally known for their high risk of asbestos exposure, showed high SIRs in this study. In addition, a high risk was also found in the accommodation and food service activities (men 2.56, 95% CI: 1.28–4.58, women 1.354, 95% CI: 0.65–2.48) and real estate (men 1.34, 95% CI: 0.98–1.83, women 1.95, 95% CI: 0.78–4.02). Even after asbestos was banned, the risk of asbestos-containing materials in old buildings still exists. Moreover, the incidence of malignant mesothelioma is likely to increase in the future, given the long latency of this disease. This study had limitations because the cohort was based on the employment insurance database. Therefore, further long-term follow-up studies will be needed to expand the cohort’s inclusiveness in the future and analyze the occupations and industries.

## Figures and Tables

**Figure 1 ijerph-18-10682-f001:**
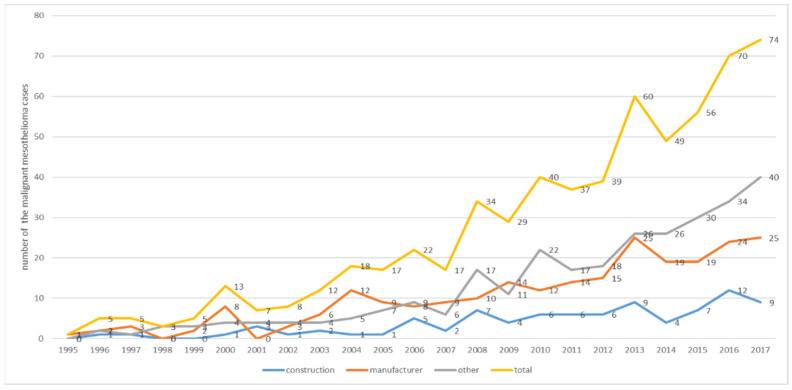
Number of malignant mesothelioma cases by year.

**Table 1 ijerph-18-10682-t001:** General characteristics of the cohort.

	Men (%)		Women (%)		Total (%)	
Gender	13,285,895	(56.0)	10,452,705	(44.0)	23,738,600	(100.0)
Age at enrollment						
<20 yrs	463,532	(3.5)	505,326	(4.8)	968,858	(4.1)
20–50 yrs	10,004,635	(75.3)	7,860,565	(75.2)	17,865,200	(75.3)
≥50 yrs	2,817,728	(21.2)	2,086,814	(20.0)	4,904,542	(20.7)
Hiring year						
~2000	2,143,106	(16.1)	1,056,479	(10.1)	3,199,585	(13.5)
2001–2010	3,896,322	(29.3)	2,985,047	(28.6)	6,881,369	(29.0)
2011~	7,246,467	(54.5)	6,411,179	(61.3)	13,657,646	(57.5)
Average year of follow-up	8		7			
Person-Years	107,138,561		72,794,387		179,932,948	
Total death cases	448,843		85,986		534,829	
SMR of all cause death (95% CI)	1.28 (1.28–1.29)		0.95 (0.94–0.95)			
Mesothelioma						
Incidence	480		141		621	
death	216		54		270	
Longest working Industry						
A	168,355	(1.3)	68,750	(0.7)	237,105	(1.0)
B	54,097	(0.4)	7284	(0.1)	61,381	(0.3)
C	6,201,849	(46.7)	3,481,366	(33.3)	9,683,215	(40.8)
D	78,711	(0.6)	21,919	(0.2)	100,630	(0.4)
E	113,468	(0.9)	42,696	(0.4)	156,164	(0.7)
F	1,238,798	(9.3)	370,243	(3.5)	1,609,041	(6.8)
G	1,385,506	(10.4)	1,518,780	(14.5)	2,904,286	(12.2)
H	812,042	(6.1)	192,496	(1.8)	1,004,538	(4.2)
I	456,363	(3.4)	810,898	(7.8)	1,267,261	(5.3)
J	459,524	(3.5)	397,696	(3.8)	857,220	(3.6)
K	350,804	(2.6)	346,045	(3.3)	696,849	(2.9)
L	331,072	(2.5)	187,645	(1.8)	518,717	(2.2)
M	350,892	(2.6)	329,055	(3.1)	679,947	(2.9)
N	526,081	(4.0)	660,627	(6.3)	1,186,708	(5.0)
O	198,361	(1.5)	311,355	(3.0)	509,716	(2.1)
P	162,302	(1.2)	469,446	(4.5)	631,748	(2.7)
Q	201,004	(1.5)	1,032,659	(9.9)	1,233,663	(5.2)
R	62,673	(0.5)	62,860	(0.6)	125,533	(0.5)
S	124,080	(0.9)	136,157	(1.3)	260,237	(1.1)
T	156	(0.0)	965	(0.0)	1121	(0.0)
U	9555	(0.1)	3734	(0.0)	13,289	(0.1)
Asbestos Exposure						
Asbestos-JEM	1,102,008	(8.3)	1,311,577	(12.5)	2,413,585	(10.2)
Industries of asbestos-JEM	5,260,955	(39.6)	4,142,037	(39.6)	9,402,992	(39.6)

JEM: job exposure Metrix, SMR: standardized mortality ratio, CI: confidence interval, A: agriculture, forestry, and fishing, B: mining and quarrying, C: manufacturing, D: electricity, gas, steam, and air conditioning supply, E: water supply; sewage, waste management, materials recovery, F: construction, G: wholesale and retail trade, H: transportation and storage, I: accommodation and food service activities, J: information and communication, K: financial and insurance activities, L: real estate activities, M: professional, scientific and technical activities, N: business facilities management and business support services, O: Public administration and defense; compulsory social security, P: education, Q: human health and social work activities, R: arts, sports, and recreation-related services, S: membership organizations, repair, and other personal services, T: activities of households as employers, U: activities of extraterritorial organizations and bodies.

**Table 2 ijerph-18-10682-t002:** SIR and SMR for malignant mesothelioma according to the major classification of industries.

KSIC	Men			Women			Men			Women		
	OBS	SIR	95% CI	OBS	SIR	95% CI	OBS	SMR	95% CI	OBS	SMR	95% CI
Total	480	1.36	(1.24–1.48) *	141	1.44	(1.23–1.7) *	216	0.88	(0.77–1.01)	54	0.95	(0.73–1.24)
A	9	1.51	(0.69–2.86)	2	2.06	(0.25–7.45)	7	1.63	(0.66–3.37)	1	1.56	(0.04–8.69)
B	2	0.80	(0.1–2.91)	.	.		1	0.57	(0.01–3.2)	.	.	
C	183	1.54	(1.33–1.78) *	57	1.62	(1.25–2.11) *	81	1.02	(0.82–1.27)	26	1.30	(0.89–1.91)
D	5	2.02	(0.66–4.71)	.	.		1	0.62	(0.02–3.43)	.	.	
E	6	1.25	(0.46–2.72)	.	.		2	0.59	(0.07–2.12)	.	.	
F	76	1.72	(1.37–2.15) *	12	3.31	(1.71–5.79) *	43	1.40	(1.04–1.89) *	5	2.36	(0.77–5.51)
G	28	1.24	(0.85–1.79)	12	1.34	(0.69–2.35)	14	0.92	(0.5–1.54)	3	0.63	(0.13–1.84)
H	48	1.26	(0.95–1.67)	1	0.56	(0.01–3.12)	21	0.80	(0.52–1.23)	1	1.00	(0.03–5.56)
I	11	2.56	(1.28–4.58) *	10	1.35	(0.65–2.48)	6	2.07	(0.76–4.51)	4	0.91	(0.25–2.33)
J	14	1.70	(0.93–2.85)	4	1.84	(0.5–4.71)	7	1.29	(0.52–2.65)	2	1.86	(0.23–6.71)
K	6	0.49	(0.18–1.06)	4	1.48	(0.4–3.8)	1	0.12	(0–0.69)	2	1.5	(0.18–5.4)
L	39	1.34	(0.98–1.83)	7	1.95	(0.78–4.02)	13	0.60	(0.32–1.03)	3	1.26	(0.26–3.68)
M	11	1.23	(0.61–2.19)	1	0.49	(0.01–2.75)	3	0.48	(0.1–1.41)	.	.	
N	22	0.82	(0.54–1.25)	11	1.02	(0.51–1.82)	7	0.35	(0.14–0.72)	3	0.43	(0.09–1.24)
O	9	1.02	(0.47–1.94)	3	0.67	(0.14–1.95)	5	0.74	(0.24–1.73)	1	0.32	(0.01–1.81)
P	2	0.52	(0.06–1.87)	7	2.19	(0.88–4.51)	.	.		1	0.59	(0.01–3.27)
Q	3	0.58	(0.12–1.69)	7	0.92	(0.37–1.9)	2	0.55	(0.07–1.98)	2	0.45	(0.05–1.64)
R	1	0.66	(0.02–3.69)	1	1.74	(0.04–9.67)	.	.		.	.	
S	3	0.67	(0.14–1.95)	1	0.66	(0.02–3.67)	1	0.32	(0.01–1.77)	.	.	
T	.	.		1	31.4	(0.8–174.97)	.	.		.	.	
U	2	2.14	(0.26–7.73)	.	.		1	1.54	(0.04–8.59)	.	.	

*: *p* < 0.05, OBS: observations, SIR: standardized incidence ratio, SMR: standardized mortality ratio, KSIC: Korean standardized industry code, CI: confidence interval, A: agriculture, forestry and fishing, B: mining and quarrying, C: manufacturing, D: electricity, gas, steam and air conditioning supply, E: water supply; sewage, waste management, materials recovery, F: construction, G: wholesale and retail trade, H: transportation and storage, I: accommodation and food service activities, J: information and communication, K: financial and insurance activities, L: real estate activities, M: professional, scientific and technical activities, N: business facilities management and business support services, O: public administration and defense; compulsory social security, P: education, Q: human health and social work activities, R: arts, sports and recreation related services, S: membership organizations, repair and other personal services, T: activities of households as employers, U: activities of extraterritorial organizations and bodies.

**Table 3 ijerph-18-10682-t003:** SIR and SMR of malignant mesothelioma in the subdivisions of the manufacturing division.

KSIC	Men			Women			Men			Women		
	OBS	SIR	95% CI	OBS	SIR	95% CI	OBS	SMR	95% CI	OBS	SMR	95% CI
C	183	1.54	(1.33–1.78) *	57	1.62	(1.25–2.11) *	81	1.02	(0.82–1.27)	26	1.30	(0.89–1.91)
10	12	1.33	(0.69–2.33)	10	1.70	(0.81–3.12)	6	0.99	(0.36–2.15)	6	1.72	(0.63–3.74)
102	4	4.13	(1.12–10.56) *	1	0.78	(0.02–4.33)	2	3.03	(0.37–10.94)	0		
10219	2	9.60	(1.16–34.7) *	0	.		1	7.01	(0.18–39.04)	0		
103	0			3	3.97	(0.82–11.59)	0			3	6.31	(1.3–18.45) *
106	0			2	13.27	(1.61–47.94) *	0			1	11.53	(0.29–64.25)
10620	0	.		2	46.05	(5.58–166.34) *	0	.		1	43.03	(1.09–239.77) *
10713	1	1.46	(0.04–8.14)	2	4.51	(0.55–16.3)	1	2.23	(0.06–12.44)	2	8.32	(1.01–30.07) *
13	14	1.55	(0.84–2.59)	6	1.49	(0.55–3.25)	4	0.64	(0.17–1.63)	3	1.28	(0.26–3.73)
13211	3	8.66	(1.79–25.3) *	0	.		2	8.34	(1.01–30.12) *	0	.	
139	4	3.22	(0.88–8.25)	3	4.81	(0.99–14.07)	1	1.17	(0.03–6.53)	1	2.75	(0.07–15.33)
13999	3	4.49	(0.93–13.12)	3	7.27	(1.5–21.25) *	0	.		1	4.18	(0.11–23.28)
16	3	1.09	(0.23–3.19)	3	7.31	(1.51–21.36) *	2	1.05	(0.13–3.8)	1	4.11	(0.1–22.92)
162	2	1.16	(0.14–4.19)	3	10.63	(2.19–31.07) *	2	1.69	(0.2–6.11)	1	6.01	(0.15–33.48)
16232	0	.		1	62.05	(1.57–345.73) *	0	.		1	106.59	(2.7–593.9) *
17	6	1.75	(0.64–3.8)	4	5.89	(1.6–15.08) *	3	1.29	(0.27–3.76)	1	2.50	(0.06–13.95)
171	3	2.28	(0.47–6.67)	2	12.05	(1.46–43.52) *	0			0		
17124	0	.		2	102.63	(12.43–370.74) *	0	.		0	.	
17222	0	.		1	27.83	(0.7–155.06)	0	.		1	47.35	(1.2–263.79)
19	3	3.96	(0.82–11.57)	0			2	3.95	(0.48–14.28)	0		
19210	3	6.37	(1.31–18.6) *	0	.		2	6.51	(0.79–23.51)	0	.	
20	13	1.59	(0.85–2.73)	2	1.45	(0.18–5.22)	3	0.55	(0.11–1.6)	2	2.62	(0.32–9.45)
20499	4	4.28	(1.17–10.95) *	1	5.92	(0.15–32.99)	2	3.16	(0.38–11.41)	1	10.63	(0.27–59.24)
21	5	3.27	(1.06–7.62) *	0			1	0.99	(0.03–5.5)	0		
22	9	1.40	(0.64–2.65)	2	0.96	(0.12–3.45)	5	1.15	(0.37–2.69)	1	0.83	(0.02–4.61)
22232	3	10.17	(2.1–29.73) *	0	.		2	9.96	(1.21–35.99) *	0	.	
22259	2	5.60	(0.68–20.22)	0	.		2	8.24	(1–29.76) *	0		
23	10	1.31	(0.63–2.4)	2	1.85	(0.22–6.69)	3	0.57	(0.12–1.67)	1	1.56	(0.04–8.7)
22234	0	.		1	25.43	(0.64–141.71)	0	.		1	41.94	(1.06–233.66) *
24	13	1.74	(0.93–2.97)	2	3.94	(0.48–14.22)	6	1.19	(0.44–2.59)	0		
24211	1	3.29	(0.08–18.32)	1	39.87	(1.01–222.12) *	1	4.85	(0.12–27.01)	0		
25	16	1.42	(0.81–2.3)	2	1.01	(0.12–3.65)	7	0.92	(0.37–1.89)	0		
25119	4	3.78	(1.03–9.67) *	1	9.58	(0.24–53.39)	2	2.79	(0.34–10.07)	0		
26	17	2.12	(1.23–3.39) *	8	2.07	(0.89–4.08)	5	0.97	(0.32–2.27)	3	1.54	(0.32–4.5)
28	7	1.43	(0.57–2.94)	2	1.27	(0.15–4.59)	4	1.22	(0.33–3.12)	1	1.13	(0.03–6.31)
28111	2	5.57	(0.67–20.11)	0	.		2	8.18	(0.99–29.53)	0		
28902	1	40.88	(1.04–227.79) *	0	.		1	62.09	(1.57–345.96) *	0		
29	18	1.86	(1.1–2.95) *	4	2.95	(0.8–7.55)	11	1.71	(0.85–3.06)	1	1.32	(0.03–7.37)
29199	1	0.64	(0.02–3.58)	2	8.61	(1.04–31.12) *	1	0.96	(0.02–5.34)	1	7.73	(0.2–43.07)
29141	6	9.17	(3.37–19.96) *	0	.		3	6.93	(1.43–20.25) *	0		
29294	0	.		2	12.43	(1.5–44.89) *	0			0		
30	8	0.91	(0.39–1.8)	0	0	(0–3)	4	0.72	(0.2–1.85)	0		
30331	1	68.84	(1.74–383.58) *	0	.		1	102	(2.58–568.04) *	0		
31	7	1.59	(0.64–3.28)	1	2.87	(0.07–15.97)	2	0.68	(0.08–2.47)	1	5.07	(0.13–28.25)
3113	0	.		1	37.3	(0.94–207.84)	0	.		1	65.33	(1.65–363.98) *

*: *p* < 0.05, OBS: observations, SIR: standardized incidence ratio, SMR: standardized mortality ratio, KSIC: Korean standardized industry code, CI: confidence interval, C: manufacturing, 10: manufacture of food products, 102: processing and preserving of fish, crustaceans, mollusks and seaweeds, 10219: processing and preserving of other aquatic animals, 103: processing and preserving of fruit and vegetables, 106: manufacture of grain mill, starches and starch products, 10620: manufacture of starches and starch products, 10713: manufacture of sugar confectioneries and cocoa products, 13: manufacture of textiles, except apparel, 13211: weaving of cotton fabrics, 139: manufacture of other made-up textile articles, except apparel, 13999: manufacture of other textiles n.e.c., 16: manufacture of wood and of products of wood and cork; except furniture, 162:manufacture of wood products, 16232: manufacture of wooden packing boxes, drums and similar containers, 17: manufacture of pulp, paper and paper products, 171: manufacture of pulp, paper and paperboard, 17124: manufacture of lamination, composition and specific surface processing paper, 17222: manufacture of paperboard boxes and containers, 19: manufacture of coke, briquettes and refined petroleum products, 19210: petroleum refineries, 20: manufacture of chemicals and chemical products; except pharmaceuticals and medicinal chemicals, 20499: manufacture of other chemical products n.e.c., 21: manufacture of pharmaceuticals, medicinal chemical and botanical products, 22: manufacture of rubber and plastics products, 22232: manufacture of plastic cases, boxes and containers, 22259: manufacture of other foamed plastic products, 23: manufacture of other non-metallic mineral products, 23334: manufacture of concrete tiles, roofing tiles, bricks and blocks, 24: manufacture of basic metals, 24221: manufacture of copper products by rolling, extrusion and drawing, 25: manufacture of fabricated metal products, except machinery and furniture, 25119:manufacture of other structural metal products, 26: manufacture of electronic components, computer; visual, sounding and communication equipment, 28: manufacture of electrical equipment, 28111: manufacture of electric motors and generators, 28902: manufacture of electrical carbon products and insulators, 29: manufacture of other machinery and equipment, 29199: manufacture of other general-purpose machinery n.e.c., 29241: manufacture of machinery for mining, quarrying and construction, 29294: manufacture of mold and metallic patterns, 30: manufacture of motor vehicles, trailers and semitrailers, 30331: manufacture of power transmission devices for motor vehicles, 31: manufacture of other transport equipment, 31113: building of non-ferrous metal ships and other ships.

**Table 4 ijerph-18-10682-t004:** SIR and SMR of malignant mesothelioma in the subdivisions of the non-manufacturing division.

KSIC	Men			Women			Men			Women		
	OBS	SIR	95% CI	OBS	SIR	95% CI	OBS	SMR	95% CI	OBS	SMR	95% CI
F	76	1.72	(1.37–2.15) *	12	3.31	(1.71–5.79) *	43	1.40	(1.04–1.89) *	5	2.36	(0.77–5.51)
41	72	1.72	(1.37–2.17) *	12	3.54	(1.83–6.18) *	41	1.41	(1.04–1.92) *	5	2.54	(0.82–5.92)
411	25	1.63	(1.1–2.41) *	7	5.86	(2.36–12.08) *	21	1.97	(1.28–3.02) *	4	5.34	(1.45–13.67) *
41112	15	1.48	(0.83–2.44)	4	4.97	(1.36–12.74) *	14	1.99	(1.09–3.34) *	3	5.93	(1.22–17.32) *
41119	5	2.64	(0.86–6.16)	2	11.59	(1.4–41.88) *	2	1.51	(0.18–5.46)	1	9.24	(0.23–51.48)
41221	7	3.48	(1.4–7.17) *	0	.		4	2.86	(0.78–7.31)	0	.	
41999	37	1.69	(1.23–2.33) *	5	2.70	(0.88–6.3)	15	0.99	(0.55–1.63)	1	1	(0.03–5.55)
412	10	2.17	(1.04–3.99) *	0			5	1.56	(0.51–3.64)	0		
419	37	1.69	(1.23–2.33) *	5	2.70	(0.88–6.3)	15	0.99	(0.55–1.63)	1	1	(0.03–5.55)
42	4	1.63	(0.44–4.18)	0			2	1.18	(0.14–4.26)	0		
422	3	6.61	(1.36–19.33) *	0			1	3.22	(0.08–17.96)	0		
42209	2	8.62	(1.04–31.12) *				0			0		
G	28	1.24	(0.85–1.79)	12	1.34	(0.69–2.35)	14	0.92	(0.5–1.54)	3	0.63	(0.13–1.84)
46	18	1.15	(0.68–1.82)	6	1.12	(0.41–2.44)	10	0.95	(0.46–1.75)	2	0.70	(0.09–2.54)
46201	0			0			0			1	46.57	(1.18–259.44) *
46432	0	.		1	75.33	(1.91–419.73) *	0	.		0	.	
47	7	1.17	(0.47–2.41)	6	1.74	(0.64–3.79)	4	0.99	(0.27–2.53)	1	0.54	(0.01–3.01)
472	0			2	8.67	(1.05–31.31) *	0			0		
47219	0	.		2	25.35	(3.07–91.59) *	0	.		0	.	
47421	0	.		1	116.8	(2.96–650.92) *	0	.		0	.	
H	48	1.26	(0.95–1.67)	1	0.56	(0.01–3.12)	21	0.80	(0.52–1.23)	1	1	(0.03–5.56)
52	12	2.05	(1.06–3.57)	1	1.34	(0.03–7.45)	5	1.22	(0.4–2.85)	1	2.41	(0.06–13.44)
52104	1	30.89	(0.78–172.11)	0	.		1	45.01	(1.14–250.78) *	0		
I	11	2.10	(1.05–3.75) *	10	1.11	(0.53–2.04)	6	1.71	(0.63–3.72)	4	0.72	(0.2–1.85)
55	11	2.56	(1.28–4.58) *	10	1.35	(0.65–2.48)	6	2.07	(0.76–4.51)	4	0.91	(0.25–2.33)
551	7	4.03	(1.62–8.3) *	2	1.78	(0.22–6.44)	5	4.22	(1.37–9.85)	1	1.53	(0.04–8.52)
55101	5	3.45	(1.12–8.06) *	1	1.26	(0.03–7.02)	3	3.05	(0.63–8.91)	1	2.2	(0.06–12.28)
J	14	1.70	(0.93–2.85)	4	1.84	(0.5–4.71)	7	1.29	(0.52–2.65)	2	1.86	(0.23–6.71)
59	1	2.51	(0.06–13.98)	0			1	3.74	(0.09–20.82)	0		
592	1	57.81	(1.46–322.09) *	0			1	89.08	(2.26–496.31) *	0		
59201	1	71.31	(1.81–397.3) *	0	.		1	108.8	(2.75–606.1) *	0	.	
63	2	3.91	(0.47–14.14)	2	6.45	(0.78–23.28)	1	2.85	(0.07–15.9)	1	6.25	(0.16–34.84)
63999	2	6.13	(0.74–22.13)	2	9.92	(1.2–35.83) *	1	4.36	(0.11–24.29)	1	9.25	(0.23–51.53)
L	39	1.34	(0.98–1.83)	7	1.95	(0.78–4.02)	13	0.60	(0.32–1.03)	3	1.26	(0.26–3.68)
68211	31	1.4	(0.99–2)	3	1.70	(0.35–4.97)	10	0.60	(0.29–1.11)	2	1.66	(0.2–6)
O	9	1.02	(0.47–1.94)	3	0.67	(0.14–1.95)	5	0.74	(0.24–1.73)	1	0.32	(0.01–1.81)
84	9	1.02	(0.47–1.94)	3	0.67	(0.14–1.95)	5	0.74	(0.24–1.73)	1	0.32	(0.01–1.81)
842	2	1.53	(0.18–5.51)	3	6.31	(1.3–18.44) *	0			1	3.43	(0.09–19.09)
84221	0	.		1	59.51	(1.51–331.58) *	0			0		
R	1	0.66	(0.02–3.69)	1	1.74	(0.04–9.67)						
90	0			1	8.07	(0.2–44.94)	0					
90221	0	.		1	68.76	(1.74–383.12) *	0			0		

*: *p* < 0.05, OBS: observations, SIR: standardized incidence ratio, SMR: standardized mortality ratio, KSIC: Korean standardized industry code, CI: confidence interval, F: construction, 411: general construction, 41112: apartment building construction, 41119: other multi-unit house construction including apartment unit in a house, 41221: construction of highways, streets and roads, 41999: other general Construction, 412: heavy and civil engineering construction, 419: Other construction, 42: specialized construction activities, 422: construction of installing building equipment, 42209; construction of installing other building equipment, G: wholesale and retail trade, 46: wholesale trade on own account or on a fee or contract basis, 46201: wholesale of grains and oilseed crops, 46432: wholesale of electric lamps and bulbs and lighting equipment, 47: retail trade, except motor vehicles and motorcycles, 472: retail sale of foods, beverages and tobacco in specialized stores, 47219: retail sale of other foods, 47421: retail sale of household textile articles, H: transportation and storage, 52: warehousing and support activities for transportation, 52104: dangerous goods warehousing, I: accommodation and food service activities, 55: accommodation, 551: general accommodation and accommodation with cooking facilities, 55101: hotels, J: information and communication, 59: motion picture, video and television program production, sound recording and music publishing activities, 592: sound recording and music publishing activities, 59201: music and sound recordings publishing, 63: information service activities, 63999: other information service activities n.e.c., L: real estate activities, 68211: management of residential real estate, O: public administration and defense; compulsory social security, 84:, administration of industrial and social policy of community, 842: administration of general labor affairs, R: arts, sports and recreation related services, 90: creative, arts and recreation related services, 90221: museum activities.

## Data Availability

Not applicable.

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
