# Peer review of "Standardized Incidence Ratio and Standardized Mortality Ratio of Malignant Mesothelioma in a Worker Cohort Using Employment Insurance Database in Korea"

_ijerph, 2021, doi:10.3390/ijerph182010682_

Round 1
Reviewer 1 Report
The author conducted a workers' cohort using the data of employment insurance to identify the risk of malignant mesothelioma. The author showed the risk of malignant mesothelioma by industry, and suggested that there were industries with high risk of malignant mesothelioma, which were not generally considered and reported.
This study provides valuable information and insight about malignant mesothelioma in the workforce.
Although it sounds scientific, there are some points to discuss. Please consider the comments below.
<Materials and Methods section>
Please provide descriptions about the employment insurance database in the Materials and Methods section rather than Discussion section.
Please provide more detailed about the definition of disease of interest, or "malignant mesothelioma," such as ICD or KCD code used, confirmation methods by the Cancer Registration Center (histochemistry and/or markers?) and so on.
Please consider add a brief description about KSIC.
<Discussion>
The author discussed that the high SIRs of accommodation, transport and storage, and real estate industries resulted from the exposure to asbestos-containing substances of the building.
This may be one of the possible explanations, but also may be due to work experiences other than the longest work registered with employment insurance. Please discuss and explain about this possibility.
<minor comments>
There are capital letters in the middle of sentences. Please correct them or keep consistency.
Author Response
Thank you for the comments. The draft was revised according to your recommendations.

Reviewer 2 Report
I have read your manuscript, “Risk of malignant mesothelioma in a worker cohort,” and was interested in the study. However, I think that this manuscript needs a major modification.
Major comments
One of the most critical issues is that the average latency period between initial asbestos exposure and the confirmation of mesothelioma incidence is 40 years. There is a 30-50 year gap between the incidence of mesothelioma and occupational asbestos exposure. It may be difficult to consider the effect of occupational asbestos exposure as the occurrence of mesothelioma in subjects under 50 years of age. I believe there is a lack of information about the characters of the 641 malignant mesothelioma patients of subjects and the discussion regarding the incidence states of them.
Title
The title of this manuscript is too brief and may mislead the reader. The title should be more precise and informative.
Introduction
I think that author has not sufficiently described the aim of this study since the research question and research gap is vague. In this study, the author compared the incidence and mortality of mesothelioma among workers in different industries. However, the assessment of the direct asbestos exposure status of the subjects from their work is obscure. The aim of this study should be stated more clearly and precisely.
The paragraph writing of the introduction seems to be insufficient. Please reconsider the paragraph structure.
Materials and Methods
It should be presented in a structured manner, referring to the STROBE statement.
Discussion
In the discussion, there are many places where the description of results is repeated. Only the main points of the results should be described, and comparisons with previous studies and the authors' discussion should be clearly stated.
Minor comments
"Although asbestos in Korea was officially banned in 2009, its producing or using peaked since 1930's peaked in mid-1990s " seems to be a wrong sentence.
I found a lot of grammatical errors in the text. I believe that adequate proofreading is needed.
Author Response
Thank you for the detailed review. All the recommendations from the reviewer were revised in 1st revision.

Reviewer 3 Report
Please look at the comments in the manuscript.

Author Response
Thank you for your detailed and thoughtful review.
I revised the draft according to most of your comments. Please see the revised draft, the red paragraphs.

Round 2
Reviewer 2 Report
No further comments.
Author Response
The previous revision according to your comments were finished.
Thanks again for the reasonable review and comments.
Your comments have been of great help to the completion of this paper.